# Rooming-In and Breastfeeding Duration in First-Time Mothers in a Modern Postpartum Care Center

**DOI:** 10.3390/ijerph191811790

**Published:** 2022-09-19

**Authors:** Hsiao-Ling Wu, Der-Fa Lu, Pei-Kwei Tsay

**Affiliations:** 1Graduate Institute of Clinical Medical Sciences, College of Medicine, Chang Gung University, Taoyuan City 33302, Taiwan; 2Department of Nursing, Shu-Zen Junior of Medicine and Management, Kaohsiung City 82144, Taiwan; 3Eau Claire College of Nursing and Health Sciences, University of Wisconsin, Eau Claire, WI 54702-4004, USA; 4Department of Public Health and Center of Biostatistics, College of Medicine, Chang Gung University, No. 259, Wenhua 1st Rd., Guishan Dist., Taoyuan City 33302, Taiwan

**Keywords:** rooming-in, first-time mothers, exclusive breastfeeding, postpartum care center

## Abstract

Uncertainty concerning the associations between rooming-in and breastfeeding duration remains at postpartum care centers. This cross-sectional study investigated the associations between the rooming-in policy and continual exclusive breastfeeding among first-time mothers at a postpartum center. Of the 160 participants, only 12.5% (*n* = 20) implemented full rooming-in. At 3-month follow-up, 85% (*n* = 17) of those individuals were exclusively breastfeeding. At the same time point, no participant practicing partial rooming-in (*n* = 140) was exclusively breastfeeding. The generalized estimating equation analysis indicated that full (24 h) rooming-in was statistically associated with continual exclusive breastfeeding 1 month postpartum (odds ratio (OR) = 0.90, *p* < 0.001) and 3 months postpartum (OR = 0.73, *p* < 0.001). Significant factors associated with a first-time mother’s willingness to practice full rooming-in included vaginal delivery, a prenatal decision to practice breastfeeding, and undergoing prenatal classes on both rooming-in and breastfeeding. Success with continual exclusive breastfeeding in the postpartum period is dependent on full rooming-in. The findings serve as a reference for promoting exclusive breastfeeding for the first 6 months, as recommended by the World Health Organization.

## 1. Introduction

The World Health Organization (WHO) recommends that newborns should be breastfed within 1 h of birth and exclusively breastfed for the first 6 months, and that breastfeeding should be continued thereafter for ≥2 years in conjunction with complementary foods [1]. The benefits of exclusive breastfeeding during this period are well established for both mother and infant [2,3,4,5]. Despite advances in scientific knowledge related to the health benefits of breastfeeding, rates of exclusive breastfeeding are low in most parts of the world. One reason is that maintaining exclusive breastfeeding is challenging for first-time mothers [6]. The Baby-Friendly Hospital Initiative (BFHI), developed by the WHO and the United Nations Children’s Fund (UNICEF), is a global program aimed at promoting, protecting, and supporting breastfeeding [7]. The BFHI recommends the Ten Steps to Successful Breastfeeding guidelines, which emphasize the implementation of rooming-in practices [7]. Rooming-in is Step 7 in the Ten Steps to Successful Breastfeeding guidelines of the BFHI, which advises physicians to “Enable mothers and their infants to remain together and practice rooming-in throughout the day and night.” The Taiwanese government encourages hospitals to implement baby-friendly hospital practices and become certified baby-friendly hospitals [8]. In 2021, 163 maternity care providers were certified baby-friendly institutions [9]. The birth coverage rate increased from 39.2% in 2004 to 78.1% in 2017 [10]. Nevertheless, in Taiwan, the percentage of postpartum women practicing rooming-in for 24 h decreased from 27.37% in 2011 to 20.43% in 2016 [11]. Between 2012 and 2018, the rate of exclusive breastfeeding fell from 49.6% to 46.2% [12].

The postpartum period is a time of great vulnerability for many women. In Asian societies, postpartum women were traditionally provided with at-home care from nonprofessional caregivers such as their mothers or mothers-in-law for the 1–2 months following delivery. This practice is called “doing the month” [13]. In Taiwan, societal shifts away from the extended nuclear family system after the 1980s resulted in personnel at postpartum care centers largely replacing family members in administering doing-the-month care [13]. The monthly fee for a postpartum care center is between USD 5333 and USD 6666, which is four to five times higher than the average Taiwanese monthly salary [14]. The services include medical observation and consultation to mothers and infants, personal care to mothers, and dietary support according to doing-the-month customs [15]. In 2020, Taiwan’s Ministry of Health and Welfare [16] indicated that 49.19% of postpartum women were discharged to private postpartum care centers. Given that postpartum care centers follow hospitals’ lead in the implementation of rooming-in policies, a challenge that mothers face in the early postpartum period is balancing the needs of their newborns while getting adequate rest [15]. In 2018, 75% of postpartum women in South Korea used postpartum care centers. Only 3% of postpartum women practiced full rooming-in [17]. Many mothers view their stay at postpartum care centers as an opportunity to rest and recover without parenting responsibilities [15,18]. Therefore, many mothers in postpartum care centers do not desire to practice rooming-in and are separated from their babies most of the time [18,19,20].

In numerous cultures, rooming-in and nursery care are both traditional practices with their own advantages and disadvantages [7,11,20]. Ng et al. [21] found no evidence to support the premise that either practice improves breastfeeding outcomes. However, a recent systematic review has revealed the importance of continuity of care across supportive workplace policies, the Baby-Friendly Hospital Initiative, skin-to-skin care, kangaroo mother care, and cup feeding in health settings and support in community and family settings may improve breastfeeding outcomes [22]. Uncertainty concerning the associations between rooming-in and breastfeeding duration remains. In Taiwan, studies exploring the associations between rooming-in and breastfeeding duration in postpartum care centers are limited. Therefore, the present study investigated the associations between a full rooming-in policy and continual exclusive breastfeeding among first-time mothers at a postpartum care center.

## 2. Materials and Methods

### 2.1. Study Design

A cross-sectional approach was used to assess the factors related to the implementation of rooming-in practices as well as associations between continual exclusive breastfeeding and rooming-in practices. The study protocol was reviewed and approved by the Institutional Review Board of An-Nan Municipal Hospital (approval No. TMANH107-REC011).

### 2.2. Participants and Enrollment Sites

Through convenience sampling, we recruited 160 first-time mothers at a postpartum care center affiliated with a teaching hospital in southern Taiwan. The following inclusion criteria were applied: (1) being first-time mothers with full-term delivery; (2) having stayed in the postpartum care center following their discharge from the hospital; and (3) adhering to rooming-in criteria for the newborn’s health status (i.e., no medical complications, such as birth asphyxia or lack of breathing at birth). The exclusion criteria were first-time mothers with postpartum obstetric complications, such as postpartum hemorrhage.

### 2.3. Data Collection

The quantitative data were collected in September 2018. The first author contacted potential participants at the care center and invited them to join the study. The study aims and procedures were explained to the participants. After they signed the consent forms, their medical records were reviewed, and self-report questionnaires were distributed. The demographic data (i.e., age, education level, mode of delivery, parity, and duration of hospitalization) and the mode of rooming-in were retrieved using medical records.

Full-time rooming-in was defined as postnatal mothers and healthy infants staying together in the same room for 24 h a day from the time they arrive in their room after delivery. In partial rooming-in, the main care baby procedures are undertaken in the nursery room and babies spend less than 24 h in the same room as the mother. The responses were rated as “implement full-time rooming-in,” or “implement partial rooming-in.” Prenatal breastfeeding decisions and prenatal parent education programs were based on maternal self-reports. The participants were also asked if they had prenatally decided on how to feed their baby, and the responses were rated as “yes,” “not yet decided,” or “no consideration.” Prenatal parent education programs are available for expectant parents and focus on breastfeeding and caring for a newborn. Responses were rated as rooming-in guidance, feeding method guidance, or both. At 1-month and 3-month postpartum, information on breastfeeding practice status was collected. Exclusive breastfeeding was defined as giving no food or drink other than breastmilk, mixed feeding was defined as mixing breast and bottle feeding, and formula feeding was defined as feeding an infant prepared formula instead of breastfeeding.

### 2.4. Data Analysis

Analyses were performed using IBM SPSS Statistics for Windows, version 22 (IBM Corp., Armonk, NY, USA). Descriptive statistics (frequencies percentages, means, and standard deviations (SDs)) were employed to analyze the demographic data, rooming-in practice records, and breastfeeding records. Inferential statistics, specifically Fisher’s exact test and the independent samples *t*-test, were used to compare between-group differences. The associations between full versus partial rooming-in and continual exclusive breastfeeding at 1 and 3 months postpartum were explored using generalizing estimating equation (GEE) analysis. The level of significance was set to *p* < 0.005.

## 3. Results

Of the 160 participants involved in the study, 20 (12.5%) practiced full rooming-in, whereas the other 140 (87.5%) practiced partial rooming-in (Table 1). The mean age was 30.3 years (SD = 2.92), and more than 90% of the participants (*n* = 144) had a university education or higher. Regarding the mode of delivery, 77.5% of the participants (*n* = 124) delivered vaginally, and 92.5% (*n* = 148) had singleton births. The mean (SD) postpartum length of hospitalization was 3.4 (0.80) days. All participants had completed prenatal classes, with 88.1% (*n* = 141) having attended a breastfeeding class, 1.9% (*n* = 3) having attended a neonatal care class on rooming-in, and 10% (*n* = 16) having attended both types of classes. During their hospitalization, all participants had undergone full rooming-in, and 70% (*n =* 112) had practiced exclusive breastfeeding. However, by 1 month postpartum, the exclusive breastfeeding rate had declined to 12.5% (*n =* 20), with a further drop to 10.6% (*n =* 17) at 3 months postpartum. For 75.6% (*n* = 121) of the participants, the infant healthcare services provided were the main reason for staying at the care center.

The full rooming-in group (*n =* 20) had a mean age of 30.4 years (SD = 2.94), and more than 95% (*n =* 19) had a university education or higher. All participants delivered vaginally and had singleton births. The mean postpartum length of hospitalization was 3.1 (SD = 0.36) days. Furthermore, the participants in this group had received prenatal education; 5% (*n =* 1) had received a breastfeeding class, 15% (*n =* 3) had received a neonatal care course on rooming-in, and 80% (*n =* 16) had received both classes. During their hospitalization, all members of the full rooming-in group had undergone full rooming-in and practiced exclusive breastfeeding. At 1 month and 3 months postpartum, the exclusive breastfeeding rates were 100% (*n =* 20) and 85% (*n =* 17), respectively. For 55% (*n* = 11) of the participants, the infant healthcare services provided were the main reason for staying at the stay center.

The partial rooming-in group (*n =* 140) had a mean age of 30.3 (SD = 2.93) years, and more than 89.3% (*n =* 125) had a university education or higher. Regarding the mode of delivery, 74.3% (*n =* 104) delivered vaginally, and 91.4% (*n =* 128) had singleton births. The mean postpartum length of hospitalization was 3.4 (SD = 0.84) days. All the participants in this group (*n =* 140) had attended a breastfeeding class, and no member had attended a neonatal care class on rooming-in. During their hospitalization, all members of the partial rooming-in group had undergone full rooming-in, with 65.7% (*n =* 92) practicing exclusive breastfeeding. However, by 1 month and 3 months postpartum, the exclusive breastfeeding rate had fallen to 0% (*n =* 0). For 78.6% (*n* = 110) of the participants, the infant healthcare services provided were the main reason for staying at the stay center.

The mode of delivery significantly differed between participants who practiced full rooming-in at the care center and those who practiced partial rooming-in. Specifically, no one in the full rooming-in group had a cesarean section, and during the prenatal period, all members had decided to breastfeed. Moreover, this group had attended prenatal education classes on both breastfeeding and rooming-in (*p* < 0.001). Overall, 80% (*n =* 16) of the full rooming-in group had decided to breastfeed. Notably, all the full rooming-in mothers elected to breastfeed their babies exclusively. Table 2 provides a between-group comparison of exclusive breastfeeding and its links to rooming-in practices. A significant between-group difference in continual exclusive breastfeeding was observed (OR = 0.43, *p* < 0.001; 95% confidence interval 0.39–0.48). The GEE analysis of time effects revealed that full rooming-in was significantly associated with continually exclusive breastfeeding at 1 month postpartum (OR = 0.90, *p* < 0.001; 95% confidence interval 0.86–0.94) and 3 months postpartum (OR = 0.73, *p* < 0.001; 95% confidence interval 0.68–0.79).

## 4. Discussion

Overall, compared with full rooming-in, partial rooming-in resulted in a lower rate of exclusive breastfeeding. Rooming-in permits the mother to breastfeed frequently and on demand, which facilitates breastfeeding [1,2]. One reason is that when mothers and babies spend all their time together, they have abundant opportunities to “practice” breastfeeding [23]. Compared with women who are separated from their newborns, women who room-in produce more milk, produce a copious milk supply sooner, breastfeed for longer durations, and are more likely to breastfeed exclusively [24]. Herein, first-time mothers who practiced full rooming-in during hospitalization were more likely than their counterparts practicing partial rooming-in to be exclusively breastfeeding at discharge. This finding is consistent with those of Inano et al. [25], who demonstrated that the implementation of full rooming-in is the key to successful breastfeeding [25]. Rooming-in enables new mothers to learn to recognize and promptly respond to feeding cues, thus facilitating the initiation and continuation of breastfeeding [26]. Notably, a study reported that rooming-in reduced physiological stress and depressive feelings in new mothers after hospital discharge. Thus, rooming-in may help empower women in their role as mothers and in their practice of exclusive breastfeeding [20]. On the basis of the literature and the present findings, we assert that encouraging 24 h rooming-in at postpartum care centers is an effective strategy for improving breastfeeding success rates.

The BFHI, developed by the WHO and the United Nations Children’s Fund (UNICEF), is a global program aimed at promoting, protecting, and supporting breastfeeding [7]. The BFHI recommends the Ten Steps to Successful Breastfeeding guidelines, which present evidence-based hospital practices to improve breastfeeding initiation, duration, and exclusivity. In this context, the key role rooming-in plays in promoting breastfeeding initiation and continuation is widely acknowledged [7]. However, adherence to this practice by healthcare facilities is lower than that of other Steps [27]. Only 12.5% of the participants practiced full rooming-in whilst in the postpartum care center, lower than the 51.4% in 2015 corresponding rates reported in the United States [28]. In postpartum care centers in both Taiwan and South Korea, rooming-in is not the norm [17]. Many mothers do not desire to practice rooming-in during their stay at such centers and are separated from their babies most of the time [19]. Thus, the environment at postpartum care centers may be associated with lower breastfeeding success rates.

Awareness of factors influencing full rooming-in is essential to the continuation of exclusive breastfeeding among first-time mothers. Herein, the mode of delivery, prenatal breastfeeding decision, and prenatal education constituted critical determinants of rooming-in practices. Our results suggest that cesarean delivery significantly decreases the willingness to practice full rooming-in. Mothers undergoing cesarean sections experienced greater postpartum fatigue than mothers undergoing vaginal delivery. Fatigue, surgical pain, mobility difficulties, and delays in breastfeeding initiation, which are commonly experienced after cesarean delivery, can limit a mother’s ability to take care of her newborn and cause reduced suckling ability and insufficient milk supply [29,30]. Moreover, women undergoing cesarean sections balance their time and energy between infant care and their own needs, such as sleep, and recovery from cesarean delivery diverts effort from breastfeeding [29]. Cesarean delivery is therefore considered a barrier to rooming-in [31]. Therefore, postpartum care centers must be aware that mothers undergoing cesarean delivery may require higher levels of nursing care and family support to make rooming-in safe and comfortable. Such accommodations may include the provision of counseling on neonatal care and breastfeeding techniques.

Regarding other factors, consistent with the literature [32,33,34], we found that 80% of the full rooming-in group had planned during their pregnancies to practice exclusive breastfeeding. Breastfeeding intentions are a strong predictor of whether women initiate breastfeeding and implement full rooming-in. Most participants who made positive prenatal breastfeeding decisions implemented rooming-in. Consequently, to promote breastfeeding initiation, first-time mothers should receive prenatal health education.

Prenatal education constitutes an integral component of maternal prenatal care in Western countries. Prenatal education programs aim to provide pregnant women with the knowledge necessary to prepare for newborn care and breastfeeding. They also help familiarize expecting couples with the main events that will occur during pregnancy and the birthing process [35]. In Canada, the first nationwide study of women’s childbirth experiences revealed that 54.5% of Canadian women had attended prenatal classes [36]. These classes can increase maternal knowledge of indications of neonatal care, and breastfeeding during the puerperium [36,37]. Information on breastfeeding or early parenting presented during the antenatal period and assistance from specialized nurses during the early postpartum period can reinforce women’s confidence in their ability to breastfeed properly, thereby supporting the practice of full rooming-in [38,39]. Herein, maternal participation in prenatal education was significantly associated with a higher prevalence of full rooming-in [31,39]. On the basis of the literature and the present findings, we argue that encouraging prenatal education within an environment supportive of full rooming-in is an effective strategy for promoting success in exclusive breastfeeding.

According to Flacking et al., pivotal factors that facilitate attuned breastfeeding are opportunities for prolonged close physical contact with the infant, and positive relationships with and support from nurses and peers [40]. To help all women who intend to breastfeed exclusively, maternity care practices (breastfeeding within 1 h of birth, showing mothers how to breastfeed, rooming-in, breastfeeding on newborns’ demand) supportive of breastfeeding should be provided. Being at postpartum care centers is a critical time for mothers to establish exclusive breastfeeding practices. Mothers’ experiences during this time affect whether they continue to engage in exclusive breastfeeding (for a duration of their choosing) after discharge [41,42]. Around the world, there is a gap between mothers’ feeding experiences and the application of rooming-in policies [11,21]. Therefore, we recommend that postpartum care centers provide care and services supportive of rooming-in practices according to their clients’ needs and perspectives.

### Limitations

The principal limitations of this study are that the sample size was small and that all participants were recruited from a single postpartum care center. Additionally, this is a cross-sectional study and no causation can be implied. Furthermore, the findings should be confirmed with longitudinal studies or RCTs, involving larger and more diverse samples.

## 5. Conclusions

This paper enriches the literature on full rooming-in and presents evidence supporting its implementation as a means to promote continual exclusive breastfeeding. Only a small number of the participants were willing to implement full rooming-in. Therefore, we advise hospitals to overcome challenges to this practice and to enforce rooming-in-friendly policies. A simulation of neonatal care experiences (such as understanding hunger cues and how to soothe a crying baby) for first-time mothers helps to ensure a positive transition after discharge. We recommend that neonatal care simulation programs be included in prenatal classes. In addition, when advertising to or negotiating contracts with potential clients, personnel at postpartum care centers should promote rooming-in policies and avoid excessive commercial marketing emphasizing nursery care. To further enhance maternal and infant health, we recommend that future studies focus on postpartum distress among first-time mothers caused by mother–infant separation at postpartum care centers, means to provide early assistance to promote parent–child bonding, and strategies for reducing the incidence of postpartum depression.

## Figures and Tables

**Table 1 ijerph-19-11790-t001:** Characteristics of participants and comparison between partial rooming-in and full-time rooming-in participants (*n* =160).

Variables	Total Sample (*n* = 160) *Mean* ± *SD*/*n* (%)	Partial Rooming-In (*n* = 140) *Mean* ± *SD*/*n* (%)	Full-Time Rooming-In (*n* = 20) *Mean* ± *SD*/*n* (%)	*t Value/X^2^*	*p Value*
Age	30.3 ± 2.92	30.3 ± 2.93	30.4 ± 2.94	0.41	0.99 ^2^
Education					
Junior college	16 (10.0)	15 (10.7)	1 (5.0)	1.14	0.56 ^1^
University	96 (60.0)	82 (58.6)	14 (70.0)		
Master’s degree	48 (30.0)	43 (30.7)	5 (25.0)		
Childbirth method					
Vaginal birth	124 (77.5)	104 (74.3)	20 (100.0)	6.63	0.01 ^1^*
Cesarean section	36 (22.5)	36 (25.7)	0 (0.0)		
Parity					
Singleton	148 (92.5)	128 (91.4)	20 (100.0)	1.83	0.17 ^1^
Multiple birth	12 (7.5)	12 (8.6)	0 (0.0)		
Days of hospitalization	3.4 ± 0.80	3.4 ± 0.84	3.1 ± 0.36	−1.75	0.08 ^2^
Prenatal breastfeeding decision					
Yes	19 (11.8)	3 (2.1)	16 (80.0)		0.001 ^1^*
Not decided yet	138 (86.3)	134 (95.8)	4 (20.0)		
No	3 (1.9)	3 (2.1)	0 (0.0)		
Prenatal parenting education program					
Rooming-in	3 (1.9)	0 (0.0)	3 (15.0)	150.92	0.001 ^1^*
Feeding method	141 (88.1)	140 (100.0)	1 (5.0)		
Both	16 (10.0)	0 (0.0)	16 (80.0)		
Rooming-in experience during hospitalization	160 (100.0)	140 (100.0)	20 (100.0)	1.00	1.00 ^1^
Breastfeeding type					
Exclusive breastfeeding	112 (70.0)	92 (65.7)	20 (100.0)	9.76	0.007 ^1^*
Mixed breastfeeding	44 (27.5)	44 (31.4)	0 (0.0)		
Formula feeding	4 (2.5)	4 (2.9)	0 (0.0)		
Breastfeeding type (1 month after postpartum)					
Exclusive breastfeeding	20 (12.5)	0 (0.0)	20 (100.0)	13.33	0.001 ^1^*
Mixed breastfeeding	112 (70.0)	112 (80.0)	0 (0.0)		
Formula feeding	28 (17.5)	28 (20.0)	0 (0.0)		
Breastfeeding type (3 months after postpartum)					
Exclusive breastfeeding	17 (10.6)	0 (0.0)	17 (85.0)	133.53	0.001 ^1^*
Mixed breastfeeding	85 (53.1)	82 (58.6)	3 (15.0)		
Formula feeding	58 (36.3)	58 (41.4)	0 (0.0)		
Selected confinement factor					
Infant healthcare service	121 (75.6)	110 (78.6)	11 (55.0)	6.47	0.16 ^1^
Maternal care service	8 (5.0)	7 (5.0)	1 (5.0)		
Confinement service	24 (15.0)	18 (12.9)	6 (30.0)		
Equipment service	3 (1.9)	2 (1.4)	1 (5.0)		
Convenience service	4 (2.5)	3 (2.1)	1 (5.0)		

SD = Standard deviation. ^1^ Fisher’s exact test. ^2^ Independent *t*-test. * indicates significance.

**Table 2 ijerph-19-11790-t002:** Generalized estimating equations to analyze the associations between rooming-in types and exclusive breastfeeding (*n* = 160).

Variables	Odds Ratio	95% CI	*p*
Group			
Full-time rooming-in	ref		
Partial rooming-in	0.43	0.39–0.48	0.001 *
Time			
Admission	ref		
1 month after delivery	0.90	0.86–0.94	0.001 *
3 months after delivery	0.73	0.68–0.79	0.001 *

CI = Confidence interval; ref means baseline; After adjusting participants’ characteristics (childbirth method, prenatal parenting education, and feeding type). * indicates significance.

## Data Availability

The data supporting the conclusion of this article will be available upon request from the corresponding author.

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
