# Peer review of "Rooming-In and Breastfeeding Duration in First-Time Mothers in a Modern Postpartum Care Center"

_ijerph, 2022, doi:10.3390/ijerph191811790_

Round 1

Reviewer 1 Report

I thank the opportunity to have reviewed this paper, which focuses on the associations between rooming-in and exclusive breastfeeding among primipara women in Taiwan at a postpartum center. While studies focusing on the BFHI Ten Steps are highly important, the current study has important limitations.  I would encourage the authors who think thoroughly about them.

Please do not refer in any point of the study about effects or impacts.  Given the study design, only associations can be drawn.

Introduction:

Line 30: I would recommend including the full recommendations of the WHO (initiation, exclusivity and duration). As currently written only includes the WHO recommendation on exclusivity.

Line 34: I would recommend to give a brief description about BFHI before jumping ahead to Step 7.

What is the percentage of BFHI hospitals in Taiwan? Percentage of births at a BFHI certified hospital? How has this changed over time? If any of this information is available it would be very helpful to put the study in context.

Lines 46-53: As a reader with little knowledge about Taiwan’s system, it is hard to understand what a postpartum care center is. Is it a day clinic? A center were women can get breastfeeding counselling? Or do women actually stay there over night?

Postpartum care centers are private entities. Are there very expensive? Covered by health insurance?  Who accesses such centers? Without a clear context, it is hard to understand what is the extent of the contribution of the study. There seems to be a sample bias issue that needs to be addressed.

Line 59: I would suggest balancing more the conclusion that there is no evidence about rooming-in and breastfeeding outcomes. The cited study (NG et al) looks at breastfeeding duration, which is only one of different breastfeeding outcomes.  See for example: Tomori, Cecília, et al. "What works to protect, promote and support breastfeeding on a large scale: A review of reviews." Maternal & Child Nutrition (2022): e13344.

Methods:

How was the sample size decided?

Line 79: Can you say more about the definition of “no medical complications”.  This has been a subject of great debate.

In the data collection section (lines 81-87) it is unclear who signed informed consent and with what purpose, as it seems that the study used administrative secondary data form medical records.  If this is not the case please clarify.

All the variables in the data collection section need to be described.  How were they measured and operationalized? For example, how are rooming in practices being measured and defined, how are breastfeeding practices operationalized?

Results:

Based on the number of participants who practiced full-rooming in, there might be a relevant sample size issue.

The fact that those in partial rooming-in had significantly higher rates of C-sections makes the comparison very questionable, as there might be a confounding issue.

Some of the variables in Table 1 have not been defined, for example, selected confinement factor, amongst others.

Discussion:

Line 158: The sentence “Rooming-in facilitates breastfeeding” seems to have something missing.

Line 196: While mothers who had a C-section are less likely to room-in, and partial rooming-in is also associated with lower exclusively breastfeeding rates, we also need to acknowledge that lacto genesis is delayed among those having C-sections, and without adequate counseling it increases the odds of not breastfeeding exclusively. This is a methodological limitation of the analysis.

A large limitation of the study is the sample, not just the sample size.

Author Response

Responses to Reviewer 1

[General Comment] I thank the opportunity to have reviewed this paper, which focuses on the associations between rooming-in and exclusive breastfeeding among primipara women in Taiwan at a postpartum center. While studies focusing on the BFHI Ten Steps are highly important, the current study has important limitations. I would encourage the authors who think thoroughly about them.

Response: We appreciate your comments. In this study, the postpartum care centers followed the hospital system of promoting rooming-in, but few women participated in rooming-in at the centers, and few exclusively breastfed. Uncertainty concerning the effects of rooming-in on breastfeeding duration remains. Few studies from Taiwan have explored the associations between rooming-in and breastfeeding duration in postpartum care centers. Therefore, we investigated the associations between the rooming-in policy and exclusive breastfeeding among first-time mothers at a postpartum care center.

[Comment 1] Please do not refer in any point of the study about effects or impacts. Given the study design, only associations can be drawn.

Response: We have made the necessary changes accordingly.

(1) Uncertainty concerning the associations between rooming-in and breastfeeding duration remains at postpartum care centers. This study investigated the associations between the rooming-in policy and exclusive breastfeeding among first-time mothers at a postpartum center [Revision on Lines 12–14].

(2) Uncertainty concerning the associations between rooming-in and breastfeeding duration remains. In Taiwan, few studies have explored the associations between rooming-in and breastfeeding duration in postpartum care centers. Therefore, we investigated the associations between a full rooming-in policy and exclusive breastfeeding among first-time mothers at a postpartum care center [Revision on Lines 76–81].

(3) A cross-sectional approach was used to assess the factors related to the implementation of rooming-in practices as well as associations between exclusive breastfeeding and rooming-in practices [Revision on Lines 84–86].

(4) The associations between full versus partial rooming-in and exclusive breastfeeding at 1 and 3 months postpartum were explored using generalizing estimating equation (GEE) analysis [Revision on Lines 125–127].

(5) The GEE analysis of time effects revealed that full rooming-in was significantly associated with continually exclusive breastfeeding 1 month postpartum (OR = .90, p < .001) and 3 months postpartum (OR = .73, p < .001) (Revision on Lines 173–175].

(6) Table 2. Generalized estimating equations to analyze the associations between rooming-in types and exclusive breastfeeding (N = 160) [Revision on Lines 182–183].

[Comment 2] Introduction: Line 30: I would recommend including the full recommendations of the WHO (initiation, exclusivity and duration). As currently written only includes the WHO recommendation on exclusivity.

Response: Thank you for this valuable information. We have revised the introduction of this study.

The World Health Organization (WHO) recommends that newborns should be breastfed within 1 h of birth and exclusively breastfed for the first 6 months and that breastfeeding should be continued thereafter for ≥2 years in conjunction with complementary foods [1]. [Revision on Lines 30–33]

[Comment 3] Line 34: I would recommend to give a brief description about BFHI before jumping ahead to Step 7.

Response: We have added a more detailed description of BFHI before Step 7.

The Baby-Friendly Hospital Initiative (BFHI), developed by the WHO and the United Nations Children’s Fund (UNICEF), is a global program aimed at promoting, protecting, and supporting breastfeeding [7]. The BFHI recommends Ten Steps to Successful Breastfeeding, which emphasizes the implementation of rooming-in practices [7].[Revision on Lines 37–40]

[Comment 4] What is the percentage of BFHI hospitals in Taiwan? Percentage of births at a BFHI certified hospital? How has this changed over time? If any of this information is available it would be very helpful to put the study in context.

Response: In the Introduction section, we have accordingly provided data and evidence related to the percentage and trends of BFHI hospitals in Taiwan.

The Taiwanese government encourages hospitals to implement baby-friendly hospital practices and become certified baby-friendly hospitals [8]. In 2021, 163 maternity care providers were certified baby-friendly institutions [9]. The birth coverage rate increased from 39.2% in 2004 to 78.1% in 2017 [10]. In Taiwan… [11]. [Revision on Lines 43–46]

[Comment 5] Lines 46-53: As a reader with little knowledge about Taiwan’s system, it is hard to understand what a postpartum care center is. Is it a day clinic? A center were women can get breastfeeding counselling? Or do women actually stay there overnight. Postpartum care centers are private entities. Are there very expensive? Covered by health insurance? Who accesses such centers? Without a clear context, it is hard to understand what is the extent of the contribution of the study. There seems to be a sample bias issue that needs to be addressed.

Response: We agree with you and have made several changes to clarify these points. In Taiwan, following delivery, women are entitled to 3–5 days of hospital stay, which is covered by Taiwan’s National Health Insurance; after this, they are discharged to a self-pay postpartum care center, where the mother and infant might rest for 1–2 months. The monthly fee for such a center is US$ 5,333–6,666, which is four to five times higher than the average monthly salary in Taiwan. These centers typically advertise themselves as five-star hotel–like institutions with professional 24-h services in line with the traditional “doing-the-month” practice for postpartum care. The services include medical observation and consultation to mothers and infants, personal care for mothers, and dietary support according to doing-the-month customs.

The monthly fee for a postpartum care center is between US$ 5,333 and US$ 6,666, which is four to five times higher than the average Taiwanese monthly salary [14]. The services include medical observation and consultation to mothers and infants, personal care to mothers, and dietary support according to doing-the-month customs [15]. [Revision on Lines 56–59]

[Comment 6] Line 59: I would suggest balancing more the conclusion that there is no evidence about rooming-in and breastfeeding outcomes. The cited study (NG et al) looks at breastfeeding duration, which is only one of different breastfeeding outcomes. See for example: Tomori, Cecília, et al. "What works to protect, promote and support breastfeeding on a large scale: A review of reviews." Maternal & Child Nutrition (2022): e13344.

Response: Thank you for this valuable information. We have updated the references and citations in the text.

In numerous cultures, rooming-in and nursery care are traditional practices with their own advantages and disadvantages [7,11, 20]. Ng et al. [21] found no evidence to support the premise that either practice improves breastfeeding outcomes. However, a recent systematic review has revealed the importance of continuity of care across supportive workplace policies, the Baby‐Friendly Hospital Initiative, skin-to-skin care, kangaroo mother care, and cup feeding in health settings and support in community and family settings may improve breastfeeding outcomes [22]. Uncertainty concerning…[Revision on Lines 70–76]

[Comment 7] Methods: How was the sample size decided?

Response: We appreciate this comment. This was a single-center study. On the basis of approximately 180 first-time mothers in the previous year, we invited them to join the study and collected 160 first-time mothers for 1 year as the sample size.

[Comment 8] Line 79: Can you say more about the definition of “no medical complications”. This has been a subject of great debate.

Response: We have clarified the sentence.

…; and (3) adhering to rooming-in criteria for the newborn’s health status (i.e., no medical complications, such as birth asphyxia or lack of breathing at birth). [Revision on Lines 93–95]

[Comment 9] In the data collection section (lines 81-87) it is unclear who signed informed consent and with what purpose, as it seems that the study used administrative secondary data form medical records. If this is not the case please clarify.

Response: We obtained written informed consent from all participants. Detailed information is provided in the Research Ethics section.

The first author contacted potential participants at the care center and invited them to join the study. The study aims and procedures were explained to the participants. After they signed the consent forms, their medical records were reviewed, and self-report questionnaires were distributed. [Revision on Lines 98–101]

[Comment 10] All the variables in the data collection section need to be described. How were they measured and operationalized? For example, how are rooming in practices being measured and defined, how are breastfeeding practices operationalized?

Response: We have made the necessary changes.

…The demographic data (i.e., age, education level, mode of delivery, parity, and duration of hospitalization) and the mode of rooming-in were retrieved using medical records. Full-time rooming-in was defined as postnatal mothers and healthy infants staying together in the same room for 24 h a day from the time they arrive in their room after delivery. In partial rooming-in, the main care baby procedures are taken in the nursery room and not 24h but partial time in a day for them a day in the same room as the mother. The responses were rated as “implement full-time rooming-in,” or “implement partial rooming-in.” Prenatal breastfeeding decisions and prenatal parent education programs were based on maternal self-reports. The participants were also asked if they had prenatally decided on how to feed their baby, and the responses were rated as “yes,” “not yet decided,” or “no consideration.” Prenatal parent education programs are available for expectant parents and are focus on breastfeeding and caring for a newborn. Responses were rated as rooming-in guidance, feeding method guidance, or both. At 1-month and 3-month postpartum, information on breastfeeding practice status was collected. Exclusive breastfeeding was defined as giving no food or drink other than breastmilk, mixed feeding was defined as mixing breast and bottle feeding, and formula feeding was defined as feeding an infant prepared formula instead of breastfeeding. [Revision on Lines 101–118]

[Comment 11] Results: Based on the number of participants who practiced full-rooming in, there might be a relevant sample size issue. The fact that those in partial rooming-in had significantly higher rates of C-sections makes the comparison very questionable, as there might be a confounding issue.

Response: According to the Taiwan National Health Insurance reimbursement policy, women can stay at the hospital for 3 days (vaginal delivery) and 5 days (cesarean section). In Taiwan, the cesarean rate reached 37% in 2016, the second-highest globally after Italy, at 39% (Ying, Linn & Chang, 2019). Rooming-in (step 7) is the hardest step to implement for Cesarean delivery because fatigue after delivery is the most important difficulty encountered, followed by discomfort when moving after cesarean section and pain after delivery, and the traditional Chinese practices of postpartum confinement, which require new mothers to rest as much as possible to prevent adverse health consequences (Lai, Hung, Stocker, Chan, & Liu, 2015). According to the MOHW (2018), rooming-in rates were significantly higher in mothers undergoing vaginal delivery than in those undergoing cesarean delivery (10.0% vs. 5.0%) (Chen, 2018). Therefore, cesarean delivery significantly affects the willingness of first-time mothers to choose full-time rooming-in in Taiwan. Moreover, this descriptive explorative study aimed at providing important information for future rigorous control studies. The large between-group differences were due to random observation, and it is not possible to add more samples at this time.

[Comment 12] Some of the variables in Table 1 have not been defined, for example, selected confinement factor, amongst others.

Response: Thanks for your suggestion and revised accordingly.

Through convenience sampling, we recruited 160 first-time mothers at a postpartum care center affiliated with a teaching hospital in southern Taiwan. The following inclusion criteria were applied: (1) being first-time mothers with full-term delivery; (2) having stayed in the postpartum care center following their discharge from the hospital; and (3) adhering to rooming-in criteria for the newborn’s health status (i.e., no medical complications, such as birth asphyxia or lack of breathing at birth). The exclusion criteria were first-time mothers with postpartum obstetric complications, such as postpartum hemorrhage. [Revision on Lines 90–96]

[Comment 13] Discussion: Line 158: The sentence “Rooming-in facilitates breastfeeding” seems to have something missing.

Response: We have made revisions accordingly.

Rooming-in permits the mother to breastfeed frequently and on-demand, which facilitates breastfeeding [1,2]. [Revision on Lines 189–190]

[Comment 14] Line 196: While mothers who had a C-section are less likely to room-in, and partial rooming-in is also associated with lower exclusively breastfeeding rates, we also need to acknowledge that lacto genesis is delayed among those having C-sections, and without adequate counseling it increases the odds of not breastfeeding exclusively. This is a methodological limitation of the analysis.

Response: Thanks for your suggestion.

Awareness of factors influencing full rooming-in is essential to continuing exclusive breastfeeding among first-time mothers. Herein, the mode of delivery, prenatal breastfeeding decision, and prenatal education constituted critical determinants of rooming-in practices. Our results demonstrate that cesarean delivery significantly decreases the willingness to practice full rooming-in. Mothers undergoing cesarean sections experienced greater postpartum fatigue than mothers undergoing vaginal delivery. Fatigue, surgical pain, mobility difficulties, and delays in breastfeeding initiation, which are commonly experienced after cesarean delivery, can limit a mother’s ability to take care of her newborn and cause reduced suckling ability and insufficient milk supply [29, 30]. More women undergoing cesarean sections balance their time and energy between infant care and their own needs, such as sleep, and recovery from cesarean diverts effort from breastfeeding [29]. Cesarean delivery is therefore considered a barrier to rooming-in [31]. Therefore, postpartum care centers must be aware that mothers undergoing cesarean delivery may require higher levels of nursing care and family support to make rooming-in safe and comfortable. Such accommodations may include the provision of counseling and information on neonatal care and breastfeeding techniques. [Revision on Lines 218–233]

[Comment 15] A large limitation of the study is the sample, not just the sample size.

Response: The importance of full-time rooming-in in promoting breastfeeding initiation and continuation within the 10 Steps for Successful Breastfeeding is widely acknowledged. However, adherence to this practice by healthcare facilities in Taiwan's BFHI is lower than that of other Steps. When promoting rooming-in to mothers, many sociocultural and individual aspects must be considered.

According to the Ministry of Internal Affairs (MOI), Taiwan had a record low of 165,249 births in 2020 (Taiwan News, 2021). During the first six months of 2022, the birth rate was down by 9 percent from the same period in 2021 (Taiwan News, 2021). Central Intelligence Agency (CIA) (2021) statistics show that among the 227 countries and regions, Taiwan ranks last at 1.07 children per woman (Taiwan News, 2021). According to CIA statistics, most women are first-time mothers in Taiwan. First-time mothers' possible physical difficulties, fatigue, lack of experience, and emphasis on rest during confinement may be unwilling to implementation of full-time rooming-in (Consales et al., 2020).

In Taiwan, studies exploring the associations of rooming-in on breastfeeding duration in postpartum care centers are limited. Uncertainty concerning the associations of rooming-in on breastfeeding duration remains. Therefore, this study investigates the association between full-time rooming-in and exclusive breastfeeding rates in first-time mothers. This is a descriptive explorative study aimed to provide important information for future rigorous control studies.

We extend our heartfelt thanks to all reviewers for your thoughtful observations, questions, and suggestions, which have truly served to clarify and strengthen our manuscript.

Reviewer 2 Report

In the 21st century it seems surprising that it is still possible to discuss and consider the positive effects of the rooming-in system. The advantages have long been proven, tested and described.  But in the other hands there are never too few publications pointing to the advantages of breastfeeding. Creating conditions for mothers to stay with their children all the time in the rooming-in system immediately after birth, as well as in specific conditions depending on cultural, local habits (as in Taiwan) mother and child care organizations. Especially since such an organization as in Taiwan of postnatal care (gathering women in centers) is unknown in Europe or the USA.However, in the introduction, the authors explain the Taiwanese specificity and postpartum care center initiative which was designed in Taiwan in 2020.  Taiwan studies exploring the effects of rooming-in on breastfeeding duration in postpartum care centres are limited.  In 2020, Taiwan’s Ministry of Health and Welfare  indicated that  49.19% of postpartum women were discharged to private postpartum care canters this replaced the traditional  practice  of  care is called “doing the month” Such a handful of historical information is very interesting for the reviewer of this work Such a system  can depersonalized the care, distances the mother from the baby and is not conducive to lactation. Therefore, the present study investigated the effects of a full rooming-in policy on continual exclusive breastfeeding among first-time mothers (primiparas) at a postpartum care center.The author uses the a cross-sectional study first-time mothers with full-term delivery and no postpartum obstetric complications. The research methodology assumed also that the mothers   must stay in the postpartum care center following their discharge from the hospital and newborns must be healthy. All study participations had decided to breastfeed.Of the 160 participants involved in the study, 20 (12.5%) practiced full rooming-in, Only a 20 of the participants ( so  small number ) were willing to implement full rooming-in.  Detailed material characteristics are shown in Table 1 In the study very big significant differences were seen between  partial and total rooming -in group.(0.0) 17 (85.0)  as well and mode of delivery. Vaginal delivery, is conducive to full  rooming- in group.  The paper is very brief and based on one detailed and very readable and clear table. This paper enriches the literature on full rooming-in and presents evidence supporting its implementation as a means to promote continual exclusive breastfeeding. The authors in conclusions advise hospitals to overcome challenges to this practice and to enforce rooming-in- friendly policies and indicate some hints how to achieve it.Furthermore, neonatal care simulation should be covered in prenatal classes. In addition, when advertising to or negotiating contracts with potential clients, personnel at postpartum care centers should promote rooming-in policies and avoid excessive commercial marketing. In the discussion, the authors emphasize the importance of education, which is essential in understanding the advantages and necessities of breastfeeding and constant mother-child contact. The authors should clarify the method of work. The   full rooming-in and partial rooming -in system should be described in details. Of the 160 participants involved in the study, 20 (12.5%) practiced full rooming-in, 98 whereas the other 140 (87.5%) practiced partial rooming-in Such differences in group sizes also require some explanation. Why only 20 women could be recruited in full rooming in the group. References  were  properly selected and up to date. Medical as well and general  literary language is correct. Conclusion is very clear and obvious. It is so important to implementation of full rooming -in as a key to successful breastfeeding. 

Author Response

Responses to Reviewer 2

[General Comment] In the 21st century it seems surprising that it is still possible to discuss and consider the positive effects of the rooming-in system. The advantages have long been proven, tested and described. But in the other hands there are never too few publications pointing to the advantages of breastfeeding. Creating conditions for mothers to stay with their children all the time in the rooming-in system immediately after birth, as well as in specific conditions depending on cultural, local habits (as in Taiwan) mother and child care organizations. Especially since such an organization as in Taiwan of postnatal care (gathering women in centers) is unknown in Europe or the USA. However, in the introduction, the authors explain the Taiwanese specificity and postpartum care center initiative which was designed in Taiwan in 2020. Taiwan studies exploring the effects of rooming-in on breastfeeding duration in postpartum care centres are limited. In 2020, Taiwan’s Ministry of Health and Welfare indicated that 49.19% of postpartum women were discharged to private postpartum care canters this replaced the traditional practice of care is called “doing the month” Such a handful of historical information is very interesting for the reviewer of this work Such a system can depersonalized the care, distances the mother from the baby and is not conducive to lactation. Therefore, the present study investigated the effects of a full rooming-in policy on continual exclusive breastfeeding among first-time mothers (primiparas) at a postpartum care center. The author uses the a cross-sectional study first-time mothers with full-term delivery and no postpartum obstetric complications. The research methodology assumed also that the mothers must stay in the postpartum care center following their discharge from the hospital and newborns must be healthy. All study participations had decided to breastfeed. Of the 160 participants involved in the study, 20 (12.5%) practiced full rooming-in, Only a 20 of the participants ( so small number ) were willing to implement full rooming-in. Detailed material characteristics are shown in Table 1 In the study very big significant differences were seen between partial and total rooming -in group.(0.0) 17 (85.0) as well and mode of delivery. Vaginal delivery, is conducive to full rooming- in group. The paper is very brief and based on one detailed and very readable and clear table. This paper enriches the literature on full rooming-in and presents evidence supporting its implementation as a means to promote continual exclusive breastfeeding. The authors in conclusions advise hospitals to overcome challenges to this practice and to enforce rooming-in- friendly policies and indicate some hints how to achieve it.Furthermore, neonatal care simulation should be covered in prenatal classes. In addition, when advertising to or negotiating contracts with potential clients, personnel at postpartum care centers should promote rooming-in policies and avoid excessive commercial marketing. In the discussion, the authors emphasize the importance of education, which is essential in understanding the advantages and necessities of breastfeeding and constant mother-child contact. The authors should clarify the method of work. The full rooming-in and partial rooming -in system should be described in details. Of the 160 participants involved in the study, 20 (12.5%) practiced full rooming-in, 98 whereas the other 140 (87.5%) practiced partial rooming-in Such differences in group sizes also require some explanation. Why only 20 women could be recruited in full rooming in the group. References were properly selected and up to date. Medical as well and general literary language is correct. Conclusion is very clear and obvious. It is so important to implementation of full rooming -in as a key to successful breastfeeding.

Response: We appreciate your positive comments, which have helped immensely in improving this manuscript.

Although there are many benefits to rooming-in, it can be argued that there are also some challenges. Fully implementing the internationally approved criteria for the Baby Friendly Hospital Initiative, especially rooming-in and early skin-to-skin contact, was difficult because of traditional cultural customs and beliefs in Taiwan. According to the Ministry of Health and Welfare investigation that rooming-in rates were significantly higher in mothers who delivered vaginally than in mothers who underwent a cesarean section (10.0% vs. 5.0%) (Chen, 2018). In Taiwan, only 23.5% of postpartum women implemented the full-time rooming-in in the BFHI hospitals (Chen, 2018). Several mothers in a study by Tsai, Yang, & Wang (2016) reported feeling too tired to practice skin-to-skin contact and rooming-in in hospitals or clinics. Nakamura et al (2020) reported that primiparae have higher post-partum levels of anxiety than multiparae, with inexperience possibly exacerbating what already is an emotionally challenging situation for most women (Nakamura et al., 2020). These results underline the importance of when promoting rooming-in to mothers, it is important to take many aspects into consideration, both at a sociocultural and individual level. We must be providing additional guidance to first-time mothers in their transition to motherhood during implement full-time rooming-in.

In Taiwan, studies exploring the associations of rooming-in on breastfeeding duration in postpartum care centers are limited. Uncertainty concerning the associations of rooming-in on breastfeeding duration remains. This is descriptive explorative study aimed to provide important information for future rigorous control studies.  Therefore, this study investigates the association between full-time rooming-in and exclusive breastfeeding rates in first-time mothers. The large between-group differences were due to random observation, and it is not possible to add more samples at this time.

Revision on Methods and Limitations section:

(1) Full-time rooming-in was defined as postnatal mothers and healthy infants staying together in the same room for 24 h a day from the time they arrive in their room after delivery. In partial rooming-in, the main care baby procedures are taken in the nursery room and not 24h but partial time in a day for them a day in the same room as the mother. The responses were rated as “implement full-time rooming-in,” or “implement partial rooming-in [Revision on Lines 104–109].

(2) The principal limitations of this study are that the sample size was small and that all participants were recruited from a single postpartum care center. Additionally, this is a cross-sectional study and no causation can be implied. Further should be validated by larger and more diverse samples, or longitudinal or RCTs are required to confirm these findings [Revision on Lines 269–273].

We extend our heartfelt thanks to all reviewers for your thoughtful observations, questions, and suggestions, which have truly served to clarify and strengthen our manuscript.
